# Precision Medicine: Determination of Ribavirin Urinary Metabolites in Relation to Drug Adverse Effects in HCV Patients

**DOI:** 10.3390/ijms231710043

**Published:** 2022-09-02

**Authors:** Ottavia Giampaoli, Fabio Sciubba, Elisa Biliotti, Mariangela Spagnoli, Riccardo Calvani, Alberta Tomassini, Giorgio Capuani, Alfredo Miccheli, Gloria Taliani

**Affiliations:** 1NMR-Based Metabolomics Laboratory (NMLab), Sapienza University of Rome, 00185 Rome, Italy; 2Department of Environmental Biology, Sapienza University of Rome, 00185 Rome, Italy; 3Department of Clinical Medicine, Policlinico Umberto I, Sapienza University of Rome, 00161 Rome, Italy; 4Department of Occupational Medicine, Epidemiology and Hygiene, INAIL, Monte Porzio Catone, 00078 Rome, Italy; 5Fondazione Policlinico Universitario A. Gemelli IRCCS, 00168 Rome, Italy; 6Department of Chemistry, Sapienza University of Rome, 00185 Rome, Italy

**Keywords:** ribavirin (RBV), hepatitis C virus (HCV), severe liver fibrosis, ^1^H-NMR, urinary metabolites

## Abstract

The most commonly used antiviral treatment against hepatitis C virus is a combination of direct-acting antivirals (DAAs) and ribavirin (RBV), which leads to a shortened duration of therapy and a sustained virologic response until 98%. Nonetheless, several dose-related side effects of RBV could limit its applications. This study aims to measure the urinary concentration of RBV and its main metabolites in order to evaluate the drug metabolism ability of HCV patients and to evaluate the adverse effects, such as anemia, with respect to RBV metabolite levels. RBV and its proactive and inactive metabolites were identified and quantified in the urine of 17 HCV males with severe liver fibrosis using proton nuclear magnetic resonance (^1^H-NMR) at the fourth week (TW4) and at the twelfth week of treatment (EOT). Four prodrug urinary metabolites, including RBV, were identified and three of them were quantified. At both the TW4 and EOT stages, six HCV patients were found to maintain high concentrations of RBV, while another six patients maintained a high level of RBV proactive metabolites, likely due to nucleosidase activity. Furthermore, a negative correlation between the reduction in hemoglobin (Hb) and proactive forms was observed, according to RBV-triphosphate accumulation causing the hemolysis. These findings represent a proof of concept regarding tailoring the drug dose in relation to the specific metabolic ability of the individual, as expected by the precision medicine approach.

## 1. Introduction

Chronic hepatitis C virus (HCV) infection affects over 180 million people worldwide, remaining a major cause of cirrhosis and its complications, such as hepatocellular carcinoma, liver transplantation, and liver-related death [1]. Since 2011, HCV-infected patients have usually been treated with a combination of pegylated interferon (pegIFN) and ribavirin (RBV), leading to an increased therapeutic efficacy and a sustained virological response (SVR) of up to 40% being achieved [2]. Recently, the development of direct-acting antivirals (DAAs) and their integration into HCV treatment have improved SVR rates to 100% and enabled the duration of therapy to be shortened [3,4]. Despite this, the outcomes of DAA-based therapies may be negatively impacted by comorbidities, such as advanced cirrhosis [5] or specific HCV characteristics [6,7].

Nevertheless, it has been shown that adding RBV may allow the successful re-treatment of most prior DAA failures [5]. Indeed, the efficacy of RBV is high when it is used in combination with DAAs for both treatment-naïve and treatment-experienced patients with genotype 1 (G1) infection [8]. However, even though high rates of SVR can be achieved without RBV [9,10], the addition of RBV may be crucial for cirrhotic or treatment-experienced patients [11,12].

1-b-D-ribofuranosyl-1,2,4-triazole-3-carboxamide, known as RBV, is a guanosine nucleoside analogue with broad-spectrum antiviral activity that is a water-soluble prodrug. Once in the cell, RBV is phosphorylated by kinases into RBV-mono, di-, and triphosphate, which are pharmacologically active forms [13]. In the organism, RBV undergoes a transformation which produces the proactive metabolite 1,2,4-triazole-3-carboxamide (T-CONH_2_) and two inactive metabolites 1-β-D-ribofuranosyl-1,2,4-triazole-3-carboxilic acid (TR-COOH) and 1,2,4-triazole-3-carboxilic acid (T-COOH); these biotransformation products are excreted in feces at a level of about 10% and in urine at a level of 72% of the dose (10% of which remains unchanged) [14,15].

RBV also enters erythrocytes through *es*-transporters and, once transformed by kinases, RBV-triphosphate accumulates excessively, since enucleated red blood cells lack dephosphorylation enzymes [16], leading to the main side effect of RBV, hemolytic anemia. It has been shown that blood accumulation and exposure to RBV at critical and stable levels may be required to achieve SVR in HCV patients infected with genotype 1b and with a high viral load [17,18]. Yet, further studies are necessary to establish the optimal steady-state RBV concentration to enable SVR to minimize the adverse events.

Genome-wide association studies have shown that human single-nucleotide polymorphisms (SNPs) are associated with RBV-induced anemia. In particular, polymorphisms near to the inosine triphosphatase (ITPA) gene locus are predictive of anemia resulting from RBV treatment [19,20]. These studies identified ITPA deficiency as a major projective factor against RBV-induced hemolytic anemia; however, the clinical utility of the SNP is limited due to the low frequency of the protective allele in the human population [13]. Indeed, two functional variants (rs1127354 and rs7270101) in the ITPA gene that cause inosine triphosphatase (ITPase) deficiency were shown to protect against RBV-induced hemolytic anemia during the early stages of treatment [21], but these variants showed strong geographical and ethnic differences in allelic frequencies [22]. However, there is still a lack of data regarding the Italian population.

In addition, oxidative damage onset related to RBV or DAAs + RBV therapy has been highlighted [23,24].

Therefore, the administration of the appropriate RBV dose is essential to manage its several adverse reactions and clinical toxicity. Few studies have been carried out aiming to adapt and validate a routine assay for the quantification of bulk, blood, and urinary RBV, in combination or not. Analytical techniques such as radioimmunoassay [25], HPLC-UV or MS detection [26,27,28], capillary electrophoresis [29], and the square-wave adsorptive stripping voltametric method [30] have been tested for this purpose, since no standard assay for RBV concentration determination is available for routine laboratory use. Yet, there is still a lack of data on RBV’s bioavailability and excretion from large-scale clinical trials and very little information is available about RBV’s main excretion products [31]. In this study, we employed for the first time the NMR-based approach to investigate the concentration of RBV and its metabolites in the urine of HCV-patients undergoing DAAs + RBV therapy. In the field of precision medicine, NMR spectroscopy has already given promising results, showing the potential to contribute to disease diagnosis [32,33,34]. In this regard, NMR analysis has allowed us to identify pro-active and inactive urine metabolites of RBV and to characterize the individual profiles on the basis of high and low inactivation of metabolic ability. These results could lay the foundation for the improvement of the therapeutic regime through personalized medicine, reducing RBV-related toxic effects for each individual.

## 2. Results

### 2.1. Resonance Assignment

1-β-D-ribofuranosyl-1H-1,2,4-triazole-3-carboxamide (RBV) is a nucleoside analogous to guanosine. In Figure 1, the structural formulas of RBV and its three main urinary metabolites are shown. Four metabolites were identified, including the ribosylated (TR-COOH; RBV), deribosylated (T-CONH_2_; T-COOH), amidic (T-CONH_2_; RBV), and acid forms (TR-COOH; T-COOH).

In order to assess the assignment of RBV and its metabolites, two zones of the whole NMR spectrum have been taken into account:(1)Spectral region between 5.97 and 6.10 ppm (resonances of ribosyl moiety);(2)Aromatic region between 8.50 and 8.80 ppm (resonances of triazole moiety).

The assignments of RBV and its metabolites in the urine samples of patients under therapy are shown in Table 1.

Concerning the spectral region between 5.97 and 6.10 ppm (Figure 2), it was possible to observe two different ribosyl moieties, whose resonances were univocally assigned on the basis of TOCSY experiments (Figure 3). 

With regard to the triazolic ring (Figure 4), we observed three singlets in the spectral region between 8.50 and 8.80 ppm.

On the basis of literature data [35] and HSQC experiments (Figure 4), it was possible to univocally identify the RBV resonances, with the H-5 proton resonating at 8.76 ppm and the corresponding H-1′ of the ribosyl moiety resonating at 6.07 ppm.

Concerning the ribosyl moiety whose anomeric proton resonates at 6.01 ppm, on the basis of the relative integrals between this proton and triazolic ring one, it is possible to univocally assign the TR-COOH H-5 to the resonance at 8.65 ppm.

The singlet resonating at 8.53 ppm can be attributed to T-CONH_2_ on the basis of the HSQC experiment (Figure 5), since the corresponding carbon resonance (150 ppm) is closer to the one of RBV (150 ppm) than that of TR-COOH, which resonates at 149 ppm. The similarity in the carbon chemical shift indicates that the resonance structures of the two rings are rather similar and this is only possible if the carboxyl group is substituted with an amidic group. The absence of the ribosyl moiety mainly influences the proton chemical shift, which is 0.23 ppm lower compared to RBV.

We then spiked a control urine sample with different concentrations of T-COOH, T-CONH_2_, and TR-COOH (Figure 6), thus also confirming the obtained results on the basis of the ratio of integrals. RBV was not added to the urine sample, since it has already been reported elsewhere [35].

As a validation of this assignment, we recorded the reference spectra of each molecule (Appendix A).

### 2.2. Quantitative Analysis

In order to assess the drug metabolism changes during therapy, proactive and inactive forms of RBV were quantified. 

The quantitative analysis was carried out starting from the integration of TR-COOH, RBV, and T-CONH_2_. 

The T-COOH metabolite was identified in all the examined samples; however, its quantification was not possible due to the low intensity of the signal and the high overlapping of resonances. The concentrations of each metabolite, expressed as µmol/mmol of creatinine, are reported in Appendix A for treatment week four (TW4) and treatment week twelve (end of treatment, EOT).

Based on the relative amounts of urinary RBV and T-CONH_2_, 12 out of 17 patients were recognized to show the same urinary excretion at both TW4 and EOT, having ratios of RBV/T-CONH_2_ higher than 1 (phenotype 1, patient IDs 09, 12, 16, 20, 47, 48) and lower than 1 (phenotype 2, patient IDs 14, 19, 40, 42, 43, 50).

Each patient enrolled in this study experienced a reduction in Hb levels from baseline (T0) to TW4 and EOT, separately (Appendix A). 

To better investigate the phenomenon, Pearson correlation analysis was performed for TW4 and EOT separately, excluding patient ID 43, taking into account the amounts of RBV, its metabolites, and the delta variation in Hb with respect to the baseline (Table 2).

A negative correlation between the Hb fold ratio (Hb values at TW4 or EOT with respect to baseline) and the $\frac{RBV}{T-CONH_{2}}$ ratio was observed at both the TW4 and EOT stages, even though the latter correlation did not achieve the statistical significance.

In addition, a targeted metabolomic analysis was carried out considering the significant metabolites related to DAA treatment and severe liver fibrosis effects, according to a previously published work [24]. In particular, pseudouridine (PSI), hypoxanthine (Hyp), methylguanidine (MG), dimethylamine (DMA), tyrosine (Tyr), and glutamine (Gln) were associated with the severe liver fibrosis; 1-methylnicotinamide (1-MNA) and 3-hydroxy-3-methylbutyrate (3-HMB) were associated with the HCV clearance; and 3-hydroxyisobutyrate (3-HIB), 2,3-dihydroxy-2-methylbutyrate (2,3-DH-2-MB), and glycine (Gly) were associated with the DAA treatment effect. The NMR assignments of these metabolites are reported elsewhere [24], while the quantitative analysis is reported in Appendix A.

In order to investigate if the different urinary excretion of proactive and inactive forms could be associated with the abovementioned metabolites, Spearman correlation analysis was performed separately on TW4 and EOT datasets.

Regarding TW4 patients, TR-COOH was positively correlated with PSI and with RBV (Table 3).

In EOT patients, RBV was positively correlated with T-CONH_2_ and, interestingly, together with all its metabolites it positively correlated with Hyp (Table 4).

## 3. Discussion

Patients with chronic HCV infections can experience adverse reactions to DAA therapy. The severity of these adverse effects is dose-dependent. 

As is well known, the major dose-related toxicity of RBV is reversible hemolytic anemia, which occurs in up to one-third of patients and is due to the accumulation of RBV-triphosphate within erythrocytes. Since the RBV anemia dependent is fully reversible, the dosage of RBV is then reduced as a function of the hemoglobin (Hb) levels [36]. This is a unique biomarker useful for regulating the RBV dose during treatment; however, it cannot be used for patients with cardiomyopathy disease [37].

The interconversion of the forms of RBV is linked to the activity of three key enzymes: adenosine deaminase, nucleosidases, and adenosine kinases (Figure 7). Interestingly, these enzymes are all present within the cells, while adenosine deaminase is also present in the blood stream [38,39].

Our results showed that the reduction in Hb correlated with higher values of the RBV/T-CONH_2_ metabolite ratio, mainly at the TW4 stage (Table 2). The negative correlation observed at TW4 and EOT was in agreement with the absence of specific enzymes for RBV-triphosphate dephosphorylation in erythrocytes, leading to the accumulation of active molecules and, therefore, causing hemolysis. From our results, the relationship between the proactive metabolites remained unchanged throughout the therapy for 12 out of 17 patients, strongly suggesting the importance of quantifying the concentration of proactive metabolites excreted to allow for an individual optimization of the RBV dosage before a severe reduction in Hb levels. The difference in the phenotypes observed could be associated with the nucleosidase activity, since this enzyme regulates the equilibrium between RBV and T-CONH_2_ (Figure 7) [14].

Another interesting aspect of this study was the investigation of the relationship between the changes in RBV metabolite levels and the urinary metabolic biomarkers associated with hepatic fibrosis and the side effects of antiviral treatment. 

A positive correlation between urinary TR-COOH levels and PSI was observed at the TW4 stage. PSI is a post-transcriptionally modified nucleoside derived from mRNA catabolism. Since RBV is preferentially active in RNA-related metabolism [40], the observed correlation between these urinary metabolites was in agreement with a higher mRNA turnover. 

On the contrary, Hyp is the only metabolite that correlates positively with all metabolites of RBV at the EOT stage.

Hyp is an intermediate involved in the purine metabolic pathway and has been found to be associated with oxidative damage in liver fibrosis [24,41]. One of RBV’s proposed mechanisms of action affects purine metabolism, since RBV is a competitive inhibitor of the inosine monophosphate dehydrogenase (IMPDH) enzyme, which converts inosine-5-monophosphate into xanthine-5-monophosphate, leading to a depletion of guanosine-triphosphate (GTP) [13,16]. The positive correlation of Hyp levels with all forms of RBV metabolites is in agreement with the RBV molecular mechanism of action on purine metabolism, increasing the Hyp levels in patients with altered purine metabolism depending on hepatic fibrosis [24].

In conclusion, although this study has some limitations due to the small number and gender exclusivity of enrolled patients, it represents an important example of personalized management of pharmacological treatment to prevent undesired effects.

## 4. Materials and Methods

### 4.1. Study Population

From a population cohort of 130 HCV patients, 17 male patients with HCV severe liver fibrosis who had received DAAs + RBV therapy were selected for the analysis to minimize the interindividual variability. Each patient was considered at two time points: the fourth week of treatment (TW4) and the twelfth week of treatment (end of treatment, EOT).

The exclusion criteria were HIV co-infection, female sex, genotype other than 1, and an estimated glomerular filtration rate eGFR ≤ 60 mL/MIN/1.73 m^2^. The anthropometric and clinical data and the therapies of patients involved in this study are reported in Appendix A. Sustained virologic response (SVR), defined as serum HCV-RNA undetectability 12 weeks after EOT, was achieved in all patients.

The study protocol conformed to the ethical guidelines of the 1975 Declaration of Helsinki and was approved by the Institutional Review Board. All patients provided their written informed consent to participate in the study.

### 4.2. Chemicals

The standards of T-CONH_2_, T-COOH, and TR-COOH were provided by Toronto Research Chemicals (20 Martin Ross Avenue, Toronto, ON, Canada). Deuterium oxide (D_2_O, deuteration degree of 99.95%) was provided by Sigma-Aldrich (ON, Canada). 3-trymethylsilyl propionic 2,2,3,3-d_4_ acid, sodium salt (TSP, deuteration degree of 98%), was purchased from Sigma-Aldrich (Saint Louis, MO, USA).

### 4.3. Sample Preparation

For each urine sample, 2 mL aliquots were collected and centrifuged for 15 min at 3000× *g* in order to precipitate any particulate matter using an Itettich Zentrifugen (Tuttlingen, Germany). A total of 120 μL of a 10× concentrated solution of TSP used as an internal standard in phosphate buffer–D_2_O was added to 1200 μL of the centrifuged sample. The pH of urine was measured and adjusted in the pH range 6.95–7.05 by adding NaOH or HCl 0.1 N in order to eliminate changes in chemical shifts. Finally, 600 μL of each sample was stored at −80 °C. For the NMR analysis, the materials used were D_2_O (ISOTEC Stable Isotopes) with an isotopic purity of 99.96% and TSP (used as internal standard) with an isotopic purity of 98%. Both of them were purchased from Sigma Aldrich (St. Louis, MO, USA).

### 4.4. NMR Analysis

All spectra were recorded at 298 K on a Bruker AVANCE^TM^ 400 (Bruker Spectrospin Company, Milan, Italy) spectrometer equipped with a Bruker multinuclear z-gradient 5 mm probehead. Spectra were recorded operating at the proton frequency of 400.13 MHz and at the carbon frequency of 100.16 MHz, corresponding to 9.4 T. The experiments were carried out using a 90° pulse for 15 µs, employing the 2 s length presat pulse sequence in the instrumental routine.

^1^H-NMR spectra were acquired by collecting 64 scans for each spectrum and 64 k data points for an acquisition time of 5.45 s. The recycle delay was set to achieve a 13 s total acquisition time to avoid relaxation effects, as reported elsewhere [42]. TOCSY experiments were conducted with a spectral width of 6000 Hz in both dimensions, a data matrix of 8192 × 256 points, a mixing time of 110 ms, and a relaxation delay of 2 s. HSQC experiments were performed with spectral widths of 6000 and 25,500 Hz for the proton and carbon, respectively; a data matrix of 8192 × 256 points; and a recycle delay of 2 s. HMBC spectra were acquired with a spectral width of 6000 and 25,500 Hz for the proton and carbon, respectively; a data matrix of 8192 × 256 points; long-range constants J_C-H_ of 4, 8, and 12 Hz; and a recycle delay of 2 s. 

Monodimensional NMR spectra were processed and quantified using the ACD Lab 1D-NMR Manager ver. 12.0 software (Advanced Chemistry Development, Inc., Toronto, ON, Canada), whereas 2D-NMR spectra were processed using MestreC and ACD/Labs. The NMR spectra were manually phased, baseline-corrected, and referenced to the chemical shift of the TSP methyl resonance at δ 0.00. The quantification of metabolites was performed by comparing the integrals to the internal standard TSP integral. Metabolite’s concentrations were expressed as μmol/mmol creatinine, using the creatinine methylene group signal at 4.05 ppm as a normalizing factor.

### 4.5. Statistical Analysis

Pearson and Spearman correlations were employed for normal or non-normal distributions, respectively, to correlate the urinary levels of drug metabolites with those of Hb and targeted analysis metabolites. Both the analyses were carried out using Sigma plot 14 (Systat Software, Palo Alto, CA, USA). A *p*-value of 0.05 was considered as the threshold for statistical significance.

## Figures and Tables

**Figure 1 ijms-23-10043-f001:**
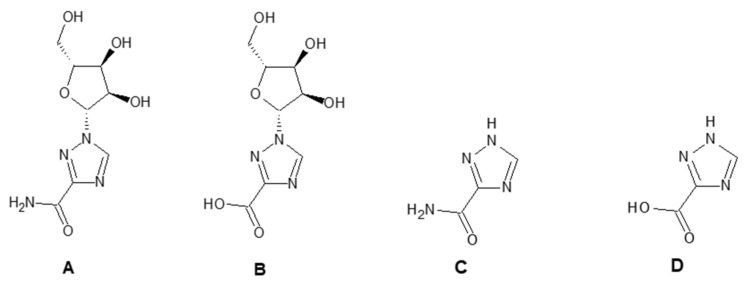
RBV and its metabolites. **A**: RBV; **B**: TR-COOH; **C**: T-CONH_2_; **D**: T-COOH.

**Figure 2 ijms-23-10043-f002:**
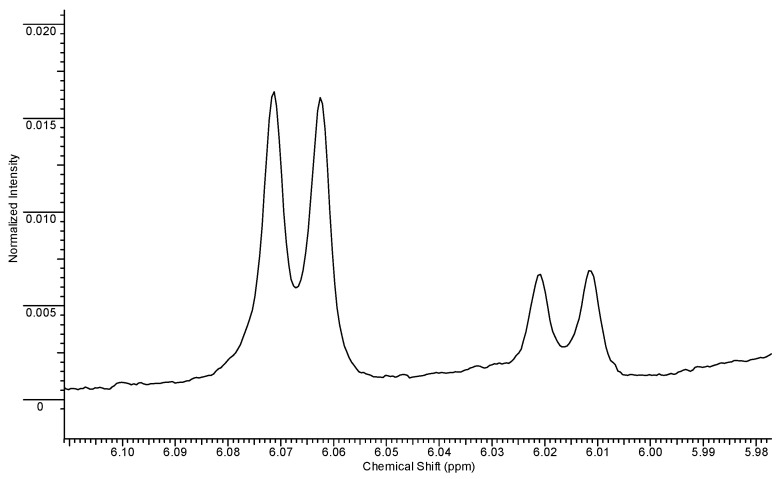
Spectral region between 5.97 and 6.10 ppm.

**Figure 3 ijms-23-10043-f003:**
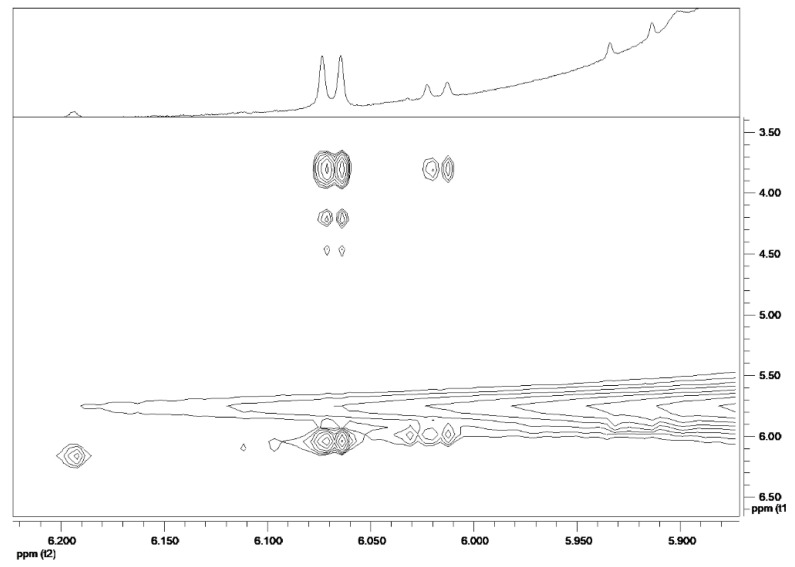
TOCSY spectrum showing correlation pattern of ribosylic moieties.

**Figure 4 ijms-23-10043-f004:**
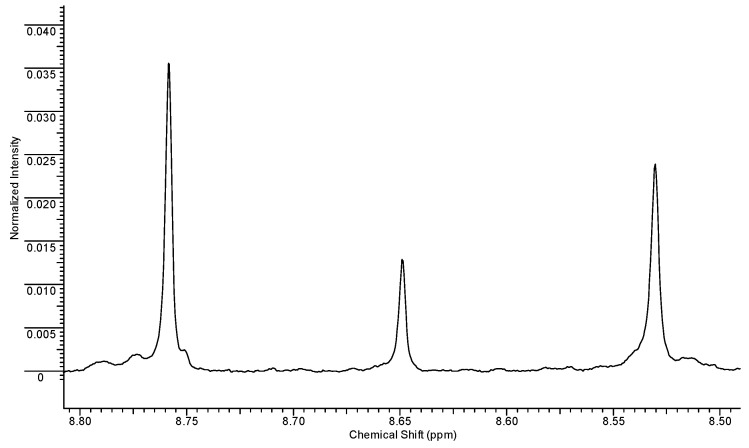
Spectral region between 8.50 and 8.80 ppm.

**Figure 5 ijms-23-10043-f005:**
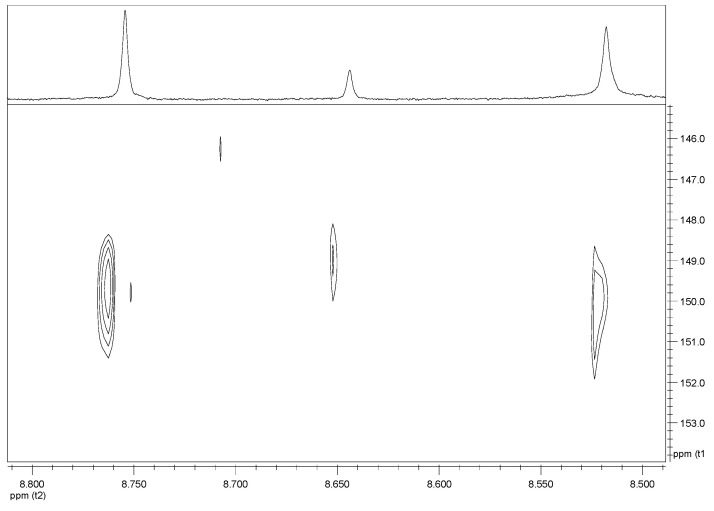
HSQC spectrum showing the resonances of triazolic H-5 for the RBV and its metabolites.

**Figure 6 ijms-23-10043-f006:**
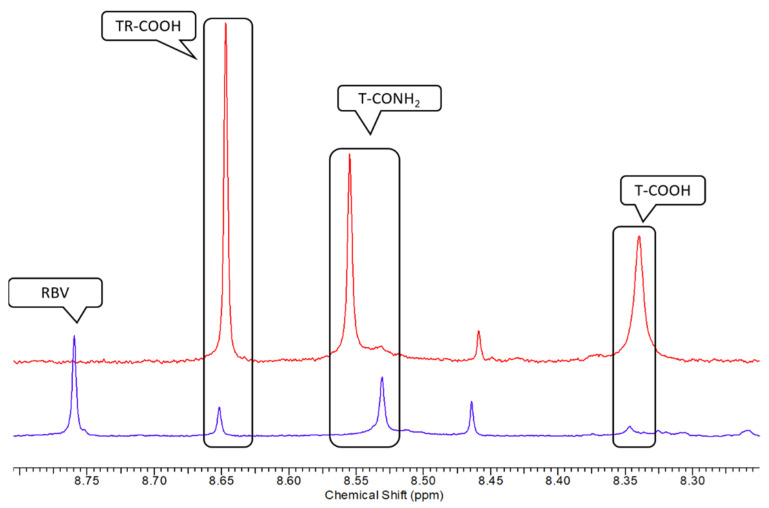
From the bottom to the top, ^1^H NMR urine aromatic region of an HCV patient at TW4 stage (blue) and a control spiked with TR-COOH, T-CONH_2_, and T-COOH (red). Small variations in chemical shift can be attributed to the different ionic strengths and the small variation in pH in the urine sample.

**Figure 7 ijms-23-10043-f007:**
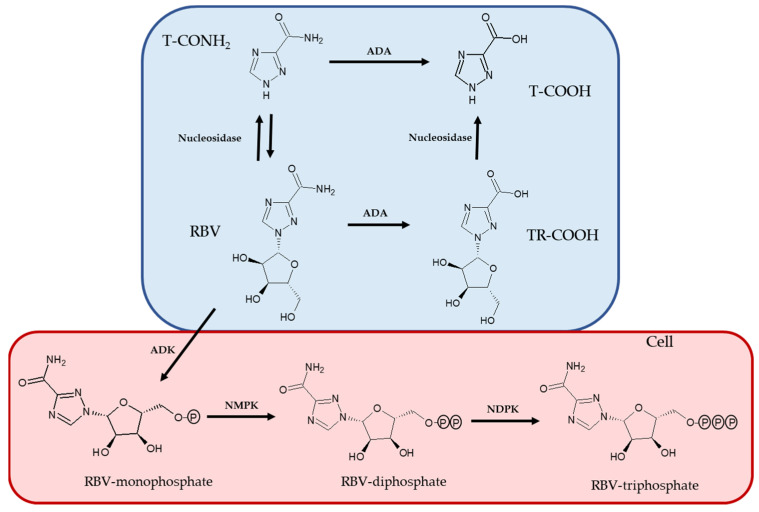
RBV metabolic pathway. Blue: extracellular compartment; red: intracellular compartment. ADA: adenosine deaminase; ADK: adenosine kinase; NMPK: nucleoside monophosphate kinase; NDPK: nucleoside diphosphate kinase.

**Table 1 ijms-23-10043-t001:** ^1^H-NMR assignment of RBV and its metabolites in urine. The integrated resonances for quantitative analysis are reported in bold.

Name	Structure	Assignment	δ (ppm) ^1^H	Multiplicity
**RBV**	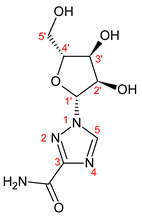	5′-CH_2_	3.80–3.90	m
		4′-CH	4.50	m
		3′-CH	4.24	m
		2′-CH	4.25	m
		1′-CH	6.07	d
		**5-CH**	**8.76**	**s**
**TR-COOH**	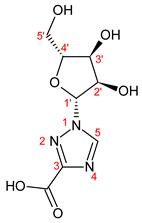	5′-CH_2_	3.80–3.90	m
		4′-CH	4.50	m
		3′-CH	4.24	m
		2′-CH	4.25	m
		1′-CH	6.01	d
		**5-CH**	**8.65**	**s**
**T-CONH_2_**	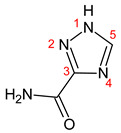	**5-CH**	**8.53**	**s**

**Table 2 ijms-23-10043-t002:** Pearson correlation values with the corresponding *p* value for TW4 and EOT patients (*n* = 16). Fold ratio of Hb was determined with respect to the baseline.

		TW4	EOT
		$\frac{RBV}{T-CONH_{2}}$	$\frac{RBV}{T-CONH_{2}}$
**TW4**	Fold Ratio Hb	**−0.520** **0.039**	
**EOT**	Fold Ratio Hb		**−0.326** **NS**

**Table 3 ijms-23-10043-t003:** Significant *p* values reported according to Spearman correlations for TW4 patients (*n* = 17).

	Correlation *p* Values		
	**T-CONH_2_**	**TR-COOH**	**RBV**
**PSI**	NS	0.036	NS
**TR-COOH**	NS	-	<0.01

**Table 4 ijms-23-10043-t004:** Significant *p* values reported according to Spearman correlations for EOT patients (*n* = 17).

	Correlation *p* Values		
	**T-CONH_2_**	**TR-COOH**	**RBV**
**Hyp**	0.038	<0.01	0.011
**T-CONH_2_**	-	<0.01	0.0164
**TR-COOH**	-	-	<0.01

## Data Availability

Data are available as Appendix A.

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
