# Peer review of "Precision Medicine: Determination of Ribavirin Urinary Metabolites in Relation to Drug Adverse Effects in HCV Patients"

_ijms, 2022, doi:10.3390/ijms231710043_

Round 1
Reviewer 1 Report (Previous Reviewer 1)
I thank authors for the correction of the manuscritp. I have no more comments.
Author Response
- We thank the reviewer for its positive comments.
Reviewer 2 Report (New Reviewer)
1. Please refer the following the articles about the serum ribavirin concentration and SVR in patients treated with interferon/ribavirin. Authors should discuss more.
Tsubota A, Arase Y, Suzuki F, Suzuki Y, Akuta N, Hosaka T, Someya T, Kobayashi M, Saitoh S, Ikeda K, Kumada H. High-dose interferon alpha-2b induction therapy in combination with ribavirin for Japanese patients infected with hepatitis C virus genotype 1b with a high baseline viral load. J Gastroenterol. 2004;39(2):155-61. doi: 10.1007/s00535-003-1266-9. PMID: 15069622
Tsubota A, Hirose Y, Izumi N, Kumada H. Pharmacokinetics of ribavirin in combined interferon-alpha 2b and ribavirin therapy for chronic hepatitis C virus infection. Br J Clin Pharmacol. 2003 Apr;55(4):360-7. doi: 10.1046/j.1365-2125.2003.01780.x. PMID: 12680884
2. Authors should more discuss about ITPA SNPs. Miyamura T, Kanda T, Nakamoto S, Wu S, Jiang X, Arai M, Fujiwara K, Imazeki F, Yokosuka O. Roles of ITPA and IL28B genotypes in chronic hepatitis C patients treated with peginterferon plus ribavirin. Viruses. 2012 Aug;4(8):1264-78. doi: 10.3390/v4081264. PMID: 23012624
How the ITPA SNPs of 12 patients? Or How the distribution of ITPA SNPs in your country??
Author Response
- We thank the reviewer for the insightful information. We addressed the suggested references in the introduction. Alas, we do not have the ITPA SNPs distribution for the patients in our study and not even for the whole Italian population.
This manuscript is a resubmission of an earlier submission. The following is a list of the peer review reports and author responses from that submission.
Round 1
Reviewer 1 Report
The work is possibly a very interesting approach to the study of drug metabolism in the human body. However, the presented results are unfortunately of dubious quality. The greatest doubts are raised by assigning chemical shifts to individual chemical compounds. Particular doubts are raised by the assignment of chemical shifts of the 5-CH protons in the triazole ring, at the same time it seems that these signals, due to the lack of overlaps, have the greatest diagnostic value. The TOSCY spectra presented do not in any way confirm the assignment of signals to individual molecules. With such similar systems, reference spectra should be recorded using the same conditions to be sure about the signal assignment.
A very quick search of the MERCK catalog shows that at least three of the four tested substances are commercially available: RVS, TR-COOH and T-COOH. The T-COONH2 molecule is probably also commercially available, at least the PUBMED database indicates several suppliers of this compound (1H-1,2,4-Triazole-3-carboxamide | C3H4N4O - PubChem (nih.gov)).
As all investigated compounds are commercially available, so authors should:
a) record the reference spectra of these compounds in aqueous solutions under the same conditions as the spectra of urine samples (temperature is an important factor)
b) register the spectra of urine samples spiked with higher and higher concentrations of individual substances. This will allow for the validation of the method of determining the concentrations used by the authors, and for determining the chemical shifts of signals characteristic for individual tested compounds.
Due to the above-mentioned reservations. I am not referring to the presented data on analyzes of patient samples, having reasonable doubts (see above) about the reliability of the presented results.
Author Response
We greatly thank the editor and the reviewers for carefully reading our manuscript and for their comments and feedbacks on this work. We modified the manuscript according to reviewers’ comments and we added the track change mode on Word file for the applied revisions.
Reviewer 1
The work is possibly a very interesting approach to the study of drug metabolism in the human body. However, the presented results are unfortunately of dubious quality. The greatest doubts are raised by assigning chemical shifts to individual chemical compounds. Particular doubts are raised by the assignment of chemical shifts of the 5-CH protons in the triazole ring, at the same time it seems that these signals, due to the lack of overlaps, have the greatest diagnostic value. The TOSCY spectra presented do not in any way confirm the assignment of signals to individual molecules. With such similar systems, reference spectra should be recorded using the same conditions to be sure about the signal assignment.
- Response: we thank the reviewer’s comments and suggestion which allowed us to greatly improve the overall quality of this manuscript. First of all, the resonance assignment was completely revaluated and entirely rewritten on the basis of new literature data and the new acquisition of 1H-13C HSQC experiments, and the changes, as well as the novel identification protocol, were added to the Results paragraph (lines 104-205). Moreover, we changed the following Discussion section according to the revised signal identification.
A very quick search of the MERCK catalog shows that at least three of the four tested substances are commercially available: RVS, TR-COOH and T-COOH. The T-COONH2 molecule is probably also commercially available, at least the PUBMED database indicates several suppliers of this compound (1H-1,2,4-Triazole-3-carboxamide | C3H4N4O - PubChem (nih.gov)).
As all investigated compounds are commercially available, so authors should:
- a) record the reference spectra of these compounds in aqueous solutions under the same conditions as the spectra of urine samples (temperature is an important factor)
- b) register the spectra of urine samples spiked with higher and higher concentrations of individual substances. This will allow for the validation of the method of determining the concentrations used by the authors, and for determining the chemical shifts of signals characteristic for individual tested compounds.
- Response: we thank the reviewer for his suggestions, but the proposed approach is mainly employed in chromatography. NMR spectroscopy is able to identify molecules on the basis of omo- and heteronuclear bidimensional experiments even without standard compounds. The quantification has been carried out by using a univocal internal standard, (trimethylsilyl)-2,2,3,3-d4 sodium propionate, in concentration equal to 2 mM, as also reported in other NMR works [https://doi.org/10.3390/ijerph19053005]. Given the nature of the molecular structures under examination, we acquired more 2D experiments and we managed to univocally assign all the resonances of interest.
Due to the above-mentioned reservations. I am not referring to the presented data on analyzes of patient samples, having reasonable doubts (see above) about the reliability of the presented results.
Reviewer 2 Report
The paper is surely of high technical quality and methodologies give convincing results. However, with the increased use of DAA in HCV treatment, RBV use is "gone although not forgotten" so that the importance of the observation seem to be low.
The authors could make some effort in the introduction and/or the discussion section to clarify to the readers the possible clinical scenarios where RBV can still get a therapeutic role (Mathur P, Kottilil S, Wilson E. Use of Ribavirin for Hepatitis C Treatment in the Modern Direct-acting Antiviral Era. J Clin Transl Hepatol. 2018;6(4):431-437. doi:10.14218/JCTH.2018.00007)and the DAA resistance issue (Di Stefano M, Faleo G, Farhan Mohamed AM, Morella S, Bruno SR, Tundo P, Fiore JR, Santantonio TA. Resistance Associated Mutations in HCV Patients Failing DAA Treatment. New Microbiol. 2021 Jan;44(1):12-18. Epub 2020 Dec 16. PMID: 33453702. ; Di Maio VC, Barbaliscia S, Teti E, Fiorentino G, Milana M, Paolucci S, Pollicino T, Morsica G, Starace M, Bruzzone B, Gennari W, Micheli V, Yu La Rosa K, Foroghi L, Calvaruso V, Lenci I, Polilli E, Babudieri S, Aghemo A, Raimondo G, Sarmati L, Coppola N, Pasquazzi C, Baldanti F, Parruti G, Perno CF, Angelico M, Craxì A, Andreoni M, Ceccherini-Silberstein F; HCV Virology Italian Resistance Network Group (Vironet C). Resistance analysis and treatment outcomes in hepatitis C virus genotype 3-infected patients within the Italian network VIRONET-C. Liver Int. 2021 Aug;41(8):1802-1814. doi: 10.1111/liv.14797. Epub 2021 Feb 8. PMID: 33497016.)
Author Response
We greatly thank the editor and the reviewers for carefully reading our manuscript and for their comments and feedbacks on this work. We modified the manuscript according to reviewers’ comments and we added the track change mode on Word file for the applied revisions.
The paper is surely of high technical quality and methodologies give convincing results. However, with the increased use of DAA in HCV treatment, RBV use is "gone although not forgotten" so that the importance of the observation seem to be low.
The authors could make some effort in the introduction and/or the discussion section to clarify to the readers the possible clinical scenarios where RBV can still get a therapeutic role (Mathur P, Kottilil S, Wilson E. Use of Ribavirin for Hepatitis C Treatment in the Modern Direct-acting Antiviral Era. J Clin Transl Hepatol. 2018;6(4):431-437. doi:10.14218/JCTH.2018.00007)and the DAA resistance issue (Di Stefano M, Faleo G, Farhan Mohamed AM, Morella S, Bruno SR, Tundo P, Fiore JR, Santantonio TA. Resistance Associated Mutations in HCV Patients Failing DAA Treatment. New Microbiol. 2021 Jan;44(1):12-18. Epub 2020 Dec 16. PMID: 33453702. ; Di Maio VC, Barbaliscia S, Teti E, Fiorentino G, Milana M, Paolucci S, Pollicino T, Morsica G, Starace M, Bruzzone B, Gennari W, Micheli V, Yu La Rosa K, Foroghi L, Calvaruso V, Lenci I, Polilli E, Babudieri S, Aghemo A, Raimondo G, Sarmati L, Coppola N, Pasquazzi C, Baldanti F, Parruti G, Perno CF, Angelico M, Craxì A, Andreoni M, Ceccherini-Silberstein F; HCV Virology Italian Resistance Network Group (Vironet C). Resistance analysis and treatment outcomes in hepatitis C virus genotype 3-infected patients within the Italian network VIRONET-C. Liver Int. 2021 Aug;41(8):1802-1814. doi: 10.1111/liv.14797. Epub 2021 Feb 8. PMID: 33497016.)
- Response: the authors greatly thank the reviewer for his positive feedback and for the comments and suggestion which allowed us to improve the quality of the manuscript. The more recent applications of RBV and the resistance toward DAAs were addressed in the introduction (lines 47-55), while, in the conclusions, the attention was focused on the generality of our approach, which could be employed in future studies regarding the metabolism of antiviral drugs.
Round 2
Reviewer 1 Report
Thanks to the authors for the corrections made to the manuscript. Unfortunately, they add nothing to the problem of identifying the tested compounds. I am surprised by the authors' statement in response to the reviews: Response: we thank the reviewer for his suggestions, but the proposed approach is mainly employed in chromatography. NMR spectroscopy can identify molecules on the basis of omo- and heteronuclear bidimensional experiments even without standard compounds”
I fully understand and I use NMR spectroscopy to identify the structure of chemical compounds. Sometimes this type of identification is possible only with NMR spectroscopy, but often other techniques (MS, IR, etc.) are needed. If the proposed method were to be used in medical diagnostics (as suggested by the title and content of the article), there can be no doubts as to the assignment of individual signals in the NMR spectrum. Unfortunately, the tested compounds are chemically very similar to each other and practically the only way to correctly assign the signals is to register the spectra of the reference compounds, which reference compounds are commercially available. The additional H1-C13 HSQC spectra recorded by the authors also do not contribute much to the identification itself (they only show correlations that are identical for all compounds and very small differences in chemical shifts). Based on the literature data from reference 30, the RSV molecule can be identified on the basis of chemical shifts, however, the identification of the remaining metabolites remains questionable.
The authors put an effort into the registration of new NMR spectra, however, these spectra as I wrote do not add anything new to the problem of correct identification of metabolites. If for some unclear reasons the authors do not want to register the reference spectra, they can possibly use appropriate spectra libraries for the identification of metabolites, e.g. Chenomx NMR Suite (Chenomx Inc., Edmonton, Canada). I have not checked whether the library of this software contains all the metabolites studied by the authors.
In summary, it supports the original opinion on this manuscript and I do not consider it fit for publication.